# Sentiment Analysis of Rumor Spread Amid COVID-19: Based on Weibo Text

**DOI:** 10.3390/healthcare9101275

**Published:** 2021-09-27

**Authors:** Peng Wang, Huimin Shi, Xiaojie Wu, Longzhen Jiao

**Affiliations:** School of Psychology, Shandong Normal University, Jinan 250358, China; wangpengpsy@sdnu.edu.cn (P.W.); Shihuiminsdnu@outlook.com (H.S.); wuxiaojiefrank2021@outlook.com (X.W.)

**Keywords:** COVID-19, rumors, sentiment analysis, topic model, time series analysis

## Abstract

(1) Background: in early 2020, COVID-19 broke out. Driven by people’s psychology of conformity, panic, group polarization, etc., various rumors appeared and spread wildly, and the Internet became a hotbed of rumors. (2) Methods: the study selected Weibo as the research media, using topic models, time series analysis, sentiment analysis, and Granger causality testing methods to analyze the social media texts related to COVID-19 rumors. (3) Results: in study 1, we obtained 21 topics related to “COVID-19 rumors” and “outbreak rumors” after conducting topic model analysis on Weibo texts; in study 2, we explored the emotional changes of netizens before and after rumor dispelling information was released and found people’s positive emotions first declined and then rose; in study 3, we also explored the emotional changes of netizens before and after the “Wuhan lockdown” event and found positive sentiment of people in non-Wuhan areas increased, while negative sentiment of people in Wuhan increased; in study 4, we studied the relationship between rumor spread and emotional polarity and found negative sentiment and rumor spread was causally interrelated. (4) Conclusion: These findings could help us to intuitively understand the impact of rumors spread on people’s emotions during the COVID-19 pandemic and help the government take measures to reduce panic.

## 1. Introduction

### 1.1. Concepts

#### 1.1.1. Rumor

As a real-world example, when the first Ebola case was diagnosed in the United States, Twitter posts mentioning Ebola jumped from 100 per min to 6000 per min and rapidly produced inaccurate claims that Ebola could be transmitted through food, water, and air [1]. Rumor is a reflection of social psychological reality [2]. Knapp [3] believes that rumor is “a widely circulated proposition related to current events, but not officially confirmed, with the purpose of making people believe”. Shibutani [4] believes that rumors are collective transactions in nature and “improvised news generated during the discussion of a group of people”. According to modern psychology, rumor refers to widespread, unproven information in a dangerous or potentially threatening situation, which is used to help understand the situation and control risk [5].

Rumor has a complex social background and psychological root [6]. From the perspective of social psychology, Allport proposed R (Rumor) = A (Ambiguous) × I (Important). Starting with mental processes, Festinger, according to cognitive dissonance theory, found that rumors do not increase anxiety but confirm it [7]. Bohm and Pfister [8] according to the cognitive and emotional dual-process model, believe the perception of the outcome of an event caused by rumors creates fear and concern. Information proliferation influences the evolution of information in two ways: by increasing competition for attention and reducing the generation time of information [9]. In addition, some researchers have found that misinformation has an advantage in competitive environments because it is freed from the constraints of being truthful, allowing it to adapt to cognition’s biases for distinctive and emotionally appealing information [10,11].

#### 1.1.2. Internet Rumor

Internet rumor refers to rumor taking Internet media as the main transmission [2]. With the development of Internet media, the variety and number of online rumors are increasing. Predicting the future development trend of rumors only through the general model lacks applicability and is prone to divergence. Therefore, it is crucial to classify Internet rumors and conduct in-depth research according to their contents and characteristics. Li et al. [12] divided rumors into conspiracy theory, expectation, and gossip from the perspective of user decision-making behavior. Peak and Hove [13] divided rumors into three categories according to event content: food safety, public security, and government behavior. Wensheng and Baihui [14] proposed a comprehensive analysis framework based on hotspot subjects, media, and publishing subjects. 

#### 1.1.3. Sentiment Analysis

Sentiment analysis refers to the automatic analysis of the relevant commentary texts content of research objects such as goods, services, and people, to find the reviewers’ positive and negative attitudes and opinions towards the research objects. Among them, sentiment classification is the most widely used, and its main task is to classify subjective texts emotionally [15]. Some studies divide emotion into positive emotion and negative emotion, while others divide emotion into positive emotion, negative emotion, and neutral emotion.

Text-based sentiment analysis is an interdisciplinary research area, involving natural language processing, databases, information retrieval, data mining, artificial intelligence, and other fields [16]. According to the different granularity of text, sentiment analysis is mainly carried out from four perspectives, including word-level, sentence-level, text-level, and topic-level sentiment analysis [17]. Generally, there are two approaches in sentimental analysis. One is by considering symbolic methods and the other one by using the machine learning method. Along with a lexicon-based and linguistic method, machine learning will be considered as one of the mainly used approaches in sentiment classification [18]. Some of the most predominant supervised learning techniques in sentiment analysis are SVM, naïve Bayesian classifiers, and other decision trees [19]. Tripathi and Naganna [20] proposed a model for sentiment analysis with movie reviews, using a combination of natural language processing and machine learning approaches. They successfully analyzed the different schemes for feature selection and their effect on sentiment analysis. The classification results clearly show that linear SVM gives more accuracy than a naïve Bayes classifier. Unsupervised learning algorithms are also known as lexicon-based techniques. Examples of unsupervised learning methods are (k means) clustering or cluster analysis [21]. In hybrid techniques, both a combination of machine learning and lexicon-based approaches are used. Researchers have proved that this combination gives improved performance of classification. Alfrjani et al. [22] presented a hybrid semantic knowledgebase machine learning approach for mining opinions at the domain feature level and classifying the overall opinion on a multi-point scale. Experimental evaluation revealed that the hybrid semantic knowledgebase machine learning approach improved the precision and recall of the extracted domain features and hence proved suitable for producing an enriched dataset of semantic features that resulted in higher classification accuracy. Dang et al. [23] applied deep learning models with TF-IDF and word embedding to Twitter datasets and implemented state-of-the-art of sentiment analysis approaches based on deep learning. Machine learning can select appropriate emotion features and use a classification method to realize emotion bias analysis [24,25].

In order to get better results for sentiment analysis, scholars have used different classifiers to study text sentiment analysis. According to the 45th China Internet development survey, as of December 2019, the number of Weibo users accounted for 42.5% of the total Internet users [26]. Difonzo [27] used tags and emoticons in Weibo as features to train KNN classifiers. Jiang [28] adopted a two-step classification method for sentiment analysis of Weibo. Guo et al. [29] analyzed classification models such as a support vector machine (SVM), naive Bayes and K-means for a Weibo corpus and used evidence theory combined with multiple classifiers to identify sentences containing opinions in Chinese Weibo. 

Sentiment analysis has a good predictive effect in many fields. Gruhl [30] predicted the peak sales ranking of bestsellers by counting the number of comments related to bestsellers in network blogs. Other scholars used relevant information on Twitter to predict the outbreak of epidemics [31], the outcome of political party elections [32], and the future development trend of the industrial index [33].

### 1.2. Characteristics and Psychological Analysis of Rumor Spread

Initial psychological studies showed that human rumor-making and rumor-spreading behaviors are mutual mappings with inner emotions, which are driven by individual emotions such as anxiety, desire, and fear [34]. Later studies pointed out that uncertainty, importance or outcome correlation, loss of control, anxiety, and belief are the five factors related to rumor spreading motivation [35]. 

As for the characteristics of rumor spread, Xiaozhe [5] divided the characteristics of rumor into situational characteristics, content characteristics, and rumor monger and audience characteristics. [36] believed that individual factors, group factors, situational factors, and psychological motivation jointly affect the spread of rumors.

Qi and Lv [37] believed that the psychological analysis of rumor spreading includes seven aspects: first, fear of the unknown catalyzes conformity; second, the collective subconscious solidifies cognitive bias; third, psychological imbalance causes negative energy release; fourth, inertia decision-making creates stereotypes; fifth, the pursuit of entertainment strengthens the psychology of curiosity; sixth, it is compensatory to meet respect needs; and seventh, subjective fluke is from punishment.

As for the social psychology behind online rumors, Zhao [38] classified them into five categories, namely, lack of security, lack of trust, accumulation and collective memory, psychological set, and group psychology specific to the group. Many rumor publishers take advantage of anonymity, which makes them lose their sense of social responsibility and self-restraint. Under the psychological control of “law does not oblige the public”, they do various acts to vent their primitive instincts and impulses [39].

### 1.3. Related Research

#### 1.3.1. Research on Internet Rumors under Public Health Emergency

Public health emergencies mainly refer to major infectious diseases, mass unexplained diseases, major food and occupational poisoning, and other events that have a serious impact on public health [40]. The evolution process of emergencies often has a life cycle [41,42,43]. Public health emergencies are characterized by sudden outbreak, complex causes, large scope and impact, high uncertainty, and the need for an emergency response [44]. Rumors are more likely to occur when the ambiguity is high and the threat or the variation is great [45].

As for the characteristics of rumor spread in public emergencies, Nie and Ma [46] believes that it mainly includes five points, namely, diversification of communication channels, fragmentation of communication content, real-time communication speed, complexity of communication motivation, and globalization of communication scope. With the advent of the era of Big Data, the Internet has become the main means of rumor generation and spread. Deng and Tang [47] divided Internet rumors into local rumors and national rumors according to the spread characteristics and influence degree of Internet rumors. Chen [48] classified Internet rumors into realistic participatory Internet rumors and original Internet rumors according to the spread path. Social media drive people’s opinions online and are the key to knowledge search, thus making digital platforms crucial for the spread of rumors. Kuang and Wu [49] classified the spread characteristics of COVID-19-related rumors into four categories: first, rumors always fluctuate with the development of the epidemic, but the overall life cycle is short; second, WeChat groups become the main channel for rumor spread; thirdly, public sentiment becomes the biggest factor affecting the spread of rumors; fourth, pictures and short videos under the banner of experts are the main forms of transmission; finally, most of the rumors focus on the prevention and treatment of COVID-19.

As for the psychological mechanisms of rumor makers, rumor mongers, and rumor refuters in public health emergencies, Du [50] believes that the psychological mechanisms of rumor makers include interest driven psychology, concerned about the chaos psychology, and deliberate destruction psychology. Rumor mongers have anxiety psychology, truth psychology, and conformity psychology, while rumor refuters have truth-seeking psychology and showing off psychology. It can be seen that the different behaviors shown by rumor makers, rumor mongers, and rumor refuters are the result of the joint action of various psychologies. As for the psychology of Internet rumor producers and mongers in the era of Big Data, Huang [51] believes that the psychological activities of people involved in the generation and dissemination of Internet rumors are becoming a kind of “self-justification” social psychology, so as to prove their “rationality” and “legitimacy” in the process of participating in Internet rumor.

#### 1.3.2. Related Research between Internet Rumors and Sentiment

Sentiment is a kind of psychological activity centered on the individual’s desire and need. It is people’s attitude experience and corresponding behavior responses to objective things. Sentiment itself belongs to neutral words, but different emotional colors endow different emotional words with sentiment polarity. An important spread motivation of rumor mongers is emotional drive [52]. When negative sentiments accumulate in large numbers, people need to find a way to release, and spreading rumors can vent negative sentiments without any cost [53].

Today, with the continuous development of social media, after an event occurs, the information related to the event will be transmitted through the network for the first time. People will also judge the event based on the received information, past experience, and values and generate personal sentiments and express them through the network. In recent years, relevant researchers have made a series of explorations on the detection and identification of Internet rumors and achieved certain results. Qazvinian et al. [54] tested the rumors generated from Twitter and selected content features, user behavior features, and tag features. The experimental results showed that these features could effectively detect network rumors, among which content features had the best effect. Takahashi and Igata [55] analyzed the rumors generated from Twitter and found that the forwarding rate of rumors was much higher than the normal rate, and there was difference between the two rates in the distribution of keywords. Sun et al. [56] conducted rumor monitoring on social hot events in Weibo and classified rumors into four categories, event untrue, outdated information, fabricated details, and picture-text mismatch, and trained the common classifiers to detect the types of picture-text mismatch rumors. Li [57] believes that the reasons for Internet rumors caused by sentiments under public health emergencies mainly include five aspects, namely, the fact that the communication space of truth is occupied by sentiment content, the vicious circle between rumor spread and negative sentiments, negative sentiments overdraft the credibility of the party and the government, sentiment spread to a certain extent promotes the development of the event, and the value of “people” has been paid more and more attention. Zheng [53] believes that irrational thinking caused by anger may make people unable to treat the information regarding public emergencies objectively.

As for the relationship between rumor and sentiment, Prasad [58] found in a study of earthquake rumors that rumor is a highly sentiment situational group reaction of the public, which stems from uncertainty and the unverifiable situation, indicating that sentiment is an important psychological reason for rumor spread. Anthony [59] found that people with high anxiety are more likely to spread rumors by investigating the impact of anxiety on rumor spread, and Faye [60] found that the public is more likely to believe and spread rumors after a catastrophic event. Wang and Qiu [61] studied the propagation law of rumor and panic through mathematical methods and simulation and found that the parallel propagation of rumor and panic promotes each other. Li et al. [62] found that the susceptibility rate of ordinary people to rumors about the source and influence of the COVID-19 was low, while the susceptibility rate to rumors about the current situation and treatment of the epidemic was high. In addition, the anxiety of ordinary people and the degree of trust in various rumors could positively predict their willingness and behavior to respread rumors.

Internet rumors affect and even dominate the development direction of public opinions by mobilizing and stimulating group emotions, especially negative emotions, and forming emotional public opinions [63]. In a sense, the generation and diffusion of network events and their public opinions are a process of emotional mobilization [64]. Therefore, rumors also have their own sentiment and emotional mobilization logic.

### 1.4. Present Study

As COVID-19 began to spread around the world in early 2020, rumors about the virus became rife because of its highly contagious nature and because little was known about it. Among them, some false information affected people’s normal life and caused social panic. Thus, what are the categories of rumors that emerged during the epidemic of COVID-19? What is the most popular category of rumors? What are the changes in netizens’ emotions before and after rumor refutation? What is the emotional impact of the “Wuhan Lockdown” on people in Wuhan and non-Wuhan areas? What is the relationship between rumors and emotional polarity? What measures should be taken with regards to netizen emotion change?

## 2. Study Design

Figure 1 showed our research design. Firstly, the main research content of this study was selected by combining the results of previous psychological research on rumors and emotions. Study 1 explored the types and popularity of rumors in the context of COVID-19; in study 2, according to the topic obtained in the first study, the rumor “71 cases of this epidemic have been confirmed in Qingdao” was selected to explore the changes of netizens’ emotions before and after the release of rumor refuting information; study 3 analyzed the emotional changes of netizens before and after the “Wuhan Lockdown” event; Study 4 explored the relationship between rumor propagation and emotional polarity in the context of COVID-19. Combined with the results of the four studies, each study was discussed, and the final results were obtained and reasonable suggestions put forward. 

## 3. Study 1 Using the Topic Model to Explore the Types and Popularity of Rumors in Weibo Texts in the Context of COVID-19 

### 3.1. Research Methods and Procedures

First of all, in order to explore the types and prevalence of Weibo rumors in the context of COVID-19, we used “COVID-19 rumors” and “epidemic refuting rumors” as keywords to conduct a fuzzy query in Weibo and collected Weibo entries from January 2020 to January 2021 using the Python crawler method. Ren et al. [65] based on Weibo Big Data, adopted keywords related to individualism/collectivism to extract Weibo texts and constructed a psychological map of Chinese collectivism. The collected fields include the sending time, text content, sending place, and other fields. After data cleaning in the pre-processing stage, a total of 19,495 valid texts were obtained.

Secondly, we used the stop words of Harbin Institute of Technology as the basis to expand the stop words list and used the JIEBA word segmentation tool to segment words and remove stop words for all blog content. Then, according to the K-means clustering algorithm, which is carried out based on the Scikit-Learn library in Python, subject model analysis was carried out on the crawled texts according to the clustering results. Topic model is one of the text dimension reduction techniques, among which the most common one is term frequency-in-verse document frequency (TF-IDF). TF-IDF is a statistical measure reflecting how important a word is to a document in a collection or corpus. The higher the frequency of a word in a target document and the lower its frequency in other documents, the greater its importance [23]. Hofmann [66] proposed PLSI (probabilisitc latent semantic indexing) to indicate that the words of the topic are subject to the polynomial distribution of the topic. Nigam et al. [67] proposed the mixture of unigrams model to represent a document semantically. The matrix was built based on a topic model, the topic content was extracted based on the determined topic bibliography, then the topic correlation degree was calculated and the topic label extracted. Finally, the topics were summarized, categorized, and ranked.

### 3.2. Results

#### 3.2.1. Topics Related to COVID-19 Rumors—Keywords

Based on the clustering results, we carried out topic model analysis on the crawled texts and got a total of 21 topics. According to the keywords given by each topic, the 21 topics are named and classified into primary and secondary categories. 

The topics of the first level classification include “Rational treatment of rumors”, “Rumor refutation information”, “World COVID-19”, “COVID-19’s origin”, “COVID-19 vaccination”, “Rumor information”, “Rumors cause social panic”, “Nucleic acid detection of suspected personnel”, and “Epidemic prevention and control”. Among them, “Rumor refutation information” includes “Company rumor refutation”, “Internet rumor refutation”, “Netizens hope the official refute rumors”, and “Rumor refutation all over China”; “Rumor information” includes “The public security bureau detained rumor mongers”, “Series of rumors of tea and garlic wipe out COVID-19 virus”, “Nine latest rumors”, “Market enterprise downtime”, “Internet spread rumors”, “Netizens spread false information”, “Weibo users release rumors”, “COVID-19 rumors in Wuhan”, “COVID-19 rumors lead to many deaths”, and “British 5 g base station spreads COVID-19 virus”.

In Table 1, except for the search keywords “COVID-19 rumor” and “Epidemic refutation”, the top six words in each topic are the keywords of each topic. 

#### 3.2.2. Ranking Topics by Popularity

We took the number of Weibo texts under each topic as the basis for judging the prevalence of rumors under that topic and ranked the topics. The results are shown in Table 2.

From the table below, we can see that in the context of COVID-19, netizens have a strong demand for official refuting of online rumors. 

## 4. Study 2 the Changes of Netizens’ Sentiments before and after the Release of Rumor Refuting Information

### 4.1. Research Purposes

In order to study whether rumor refuting information can reduce people’s negative sentiments, study 2 selected “71 cases of this epidemic have been confirmed in Qingdao” as the research object.

On 12 October 2020, a rumor titled “71 cases of this epidemic have been confirmed in Qingdao” spread widely on the Internet. On the evening of 12 October 2020, the official WeChat account “Qingdao Rumor Refuting Platform” promptly released information to refute the rumor. Therefore, we selected 13 October as the time node to explore the changes of netizens’ sentiments before and after the release of rumor refuting information.

### 4.2. Methods

#### 4.2.1. Word Embedding Model

Word embedding, also known as distributed word representation, can capture both the semantic and syntactic information of words from a large unlabeled corpus [68]. A commonly used word embedding system is Word2vec, which contains models such as the skip-gram and continuous bag-of-words (CBOW). Some of the most widely used tools for building word vectors are the models described in [69], implemented in the Word2vec tool, in particular the skip-gram and the continuous bag-of-words models. Embeddings built using the Word2vec model have been shown to capture semantic information between words, and pretraining using these models has been shown to lead to major improvements in many tasks [70]. This research mainly uses the continuous bag-of-words model of Word2vec, which was established by the team led by Tomas Mikolov [69]. Word2vec analyzes and processes words based on context to achieve the purpose of emotional orientation classification [71].

How was the classification done? First, based on the continuous bag-of-words model, the original data set was read, word segmentation and preprocessing were carried out, and the data set of the list of list format was generated. Then, the training set and test set were generated in a 7:3 ratio. Third, the program initialized the Word2vec model, specified vector dimensions, and built vocabulary. Fourth, the model was built on the comment training set to generate the word vector matrix of all the terms corresponding to the whole sentence, and the corresponding vector of the whole sentence was directly averaged by each word vector. Fifth, we generated the modeling matrix and used the transformed matrix to fit the support vector machine (SVM), k-nearest neighbors (KNN), decision tree (DT), random forest (RF) and gradient boosting machine (GBM) models, respectively, to get the classification of negative, neutral, and positive emotional texts.

#### 4.2.2. Sentiment Analysis

Sentiment analysis, also known as point of view recognition and opinion mining, refers to the analysis process of identifying, extracting, classifying, inducting, and reasoning about points, sentiment polarity, subjectivity and objectivity in the text. Xu et al. [72] analyzed the status of the emotional classification, determined a classification system, and constructed affective lexicon ontology which synthesizes various resources, which provides the basis for sentiment classification at paragraph and discourse level.

This paper analyzed and classified the emotional tendency of the words in Weibo texts and divided individuals’ emotions towards a rumor into positive, negative, and neutral categories.

Firstly, the emotional value of each sentence in the texts were calculated.
(1)$$sentenceSentimentScore=\left(\sum_{i=1}^{m}SentimentScore_{i}+\sum_{j=1}^{n}rawScore_{j}\right)\frac{1}{m\cdot n}$$

*m* indicates the number of emotional words modified by dependency, *n* indicates the number of emotional words not modified by dependency, *SentimentScore_i_* indicates the final score of the *i*-th dependency, and *rawScore_j_* indicates the emotional value of the *j*-th unmodified emotional word.

Then, through the weighted sum of the emotional value of the key sentence, the average value obtained is the emotional tendency value of the texts.
(2)$$Z=\frac{1}{k}\overset{K}{\underset{i=1}{\sum }}f\left(s_{i}\right)\cdot sentenceSentimentScore_{i}$$

*m* indicates the number of key sentences, and *f* (*si*) indicates the weight value of emotion.

Equations (1) and (2) are used to label the sentences of the training and test sets so that the sentiment analysis method used can be applied and validated later.
(3)$$\left\{\begin{array}{c} Z<\beta \\ \beta \leq Z\leq \alpha \\ Z>\alpha \end{array}\right.$$

Finally, *α* and *β* were set as the threshold values of positive and negative meanings and *Z* was set as neutral meanings, and the values of emotional inclination were normalized between 0 and 1. The emotional tendency of the whole texts can be judged by classifying the emotional value.

#### 4.2.3. Time Series Analysis

Time series analysis can be divided into deterministic change analysis and stochastic change analysis. The random change analysis methods mainly include the autoregressive model (AR), moving average model (MA), autoregressive average moving model (ARMA), and autoregressive integrated moving average model (ARIMA). The ARIMA model is a time series prediction model for random change analysis, including AR, I, and MA. AR represents an autoregressive model, I (integration) represents a single integral order, and MA represents a moving average model. In this work, the ARIMA is mainly used to analyze the obtained rumors and observe the changes of netizens’ sentiment before and after the release of rumor refuting information. Time series analysis can explore the causal relationship between sentiment value and rumors.

Time series forecasting is an important area of forecasting in which past observations of the same variable are collected and analyzed to develop a model describing the underlying relationship. Autoregressive integrated moving average (ARIMA) is one of the popular linear models in time series forecasting in the past three decades [73]. ARIMA models have been already applied to forecast commodity prices, such as oil [74]. Ariyo et al. [75] presented an extensive process of building a stock price predictive model using the ARIMA model. The results obtained from real-life data demonstrated the potential strength of ARIMA models to provide investors with short-term predictions that could aid investment decision making process. Benvenuto et al. [76] performed ARIMA model prediction on Johns Hopkins epidemiological data to predict the epidemiological trend of the prevalence and incidence of COVID-2019.

### 4.3. Results

According to the rumor refuting information released by “Qingdao Rumor Refuting Platform” about “71 cases of this epidemic have been confirmed in Qingdao”, we took 13 October as the time node to explore the emotional changes of netizens before and after the rumor refuting information was released.

Figure 2 shows the variation trend of the proportion of positive and negative texts in the total texts. From the figure below, we can clearly see that the proportion of positive texts was always higher than that of negative texts, indicating that the netizens were optimistic and positive about this rumor on the whole, and the trend of positive texts was obviously rising, while the trend of negative texts rose first and then decreased. On 13 October, as the time node of rumor refuting, the positive and negative emotion of netizens began to change, indicating that the mood changes of netizens were developing towards a positive direction and the rumor refuting work had achieved obvious results. 

According to the above results, we further explored the rumor refuting before and after the changes in the emotional value of netizens. We assign a value of 1 to positive texts, 0 to neutral texts, and −1 to negative texts to calculate the average emotional value of each day, namely:(4)$$Average\;sentimen\;value=\frac{1\;*\;the\;a\;of\;\mathrm{positive}\;\mathrm{texts}+0\;*\;\mathrm{the}\;b\;\mathrm{of}\;\mathrm{neutral}\;\mathrm{texts}+\left(-1\right)\;*\;\mathrm{the}\;c\;\mathrm{of}\;\mathrm{negative}\;\mathrm{texts}}{the\;n\;of\;all\;texts}$$

The average daily sentiment value was calculated, and its changing trend is shown in Figure 3. As can be clearly seen from the figure below, the average daily emotion value showed a noisy upward trend, and the final positive emotion was higher than that before rumor refutation. Before 13 October, the texts people posted about the rumor were the most negative, with many expressing anger about the rumor mongering and rumor mongers. With the spread of the rumor refuting information, people’s emotion gradually changed from negative to positive. In the Weibo texts published on 15 October, people all held positive attitudes towards the rumor, that is, people believed that the real situation was not like the rumor, their environment was safe, and people’s safety needs were satisfied. 

It can be concluded from Table 3 that the proportion of positive texts = (0.555 + 0.085) * time, from Table 4 that the proportion of negative texts = 0.385 = 0.085 * time, and from Table 5 that the emotional value before and after rumor refutation = −0.86 + 0.15 * time, indicating that the emotional value increases with time, so positive emotion increased and negative emotion decreased. 

## 5. Study 3 the Sentiment Changes of Netizens Related to Rumor Spreading before and after the “Wuhan Lockdown” Event 

### 5.1. Research Purposes

Study 2 explored the sentiment changes of netizens before and after a rumor was refuted under a certain topic. In study 3, we tried to explore whether netizens’ sentiments related to rumor spreading changed before and after major events during the COVID-19 outbreak. At 10 o’clock on 23 January 2020, Wuhan announced a lockdown, which is a decision that shocked the whole country and even the whole world. 

### 5.2. Results

#### 5.2.1. The Overall Changing Trend of Netizens’ Sentiments Related to Rumors before and after “Wuhan Lockdown”

For the “Wuhan Lockdown” event, we conducted a time slice and selected the texts within 24 h before and after the official announcement of the “Wuhan Lockdown” at 10 o’clock on 23 January 2020 for sentiment analysis to observe the changes of netizens’ sentiments in Weibo before and after the announcement of the lockdown.

From Figure 4, we can see that before and after the lockdown, the amount of discussion about Wuhan area in Weibo was basically the same, but the positive and negative sentiments were greatly changed. When the lockdown was announced, positive sentiments increased and negative sentiments decreased. 

From Figure 5 and Figure 6, we can see directly that in the event of “Wuhan Lockdown”, people’s sentiment value showed an obvious upward trend, and the proportion of positive texts also gradually increased, while the proportion of negative texts showed a downward trend. 

It can be concluded from Table 6 that the proportion of positive texts of “Wuhan Lockdown” = 0.590 − 0.010 * time, from Table 7 that the proportion of negative texts of “Wuhan Lockdown” = 0.731 − 0.140 * time, and from Table 8 that the sentiment value after “Wuhan Lockdown” = −0.634 + 0.140 * time. From the results of linear regression, we can see that the positive sentiments of people in the “Wuhan Lockdown” event increased significantly, while the negative sentiments decreased significantly. 

#### 5.2.2. The Sentiment Changes of Netizens in Wuhan and Non-Wuhan Areas about the Rumor in the “Wuhan Lockdown” Event

After the overall sentiment analysis of the “Wuhan Lockdown” event, we studied the sentiment changes of people in Wuhan and non-Wuhan areas respectively. According to the crawled texts with geographic information, the texts of Wuhan area and non-Wuhan area were distinguished. Then, according to the method of calculating the average sentiment value mentioned in study 2 (see Equation (4)), a chart of the change of average sentiment value in Wuhan and non-Wuhan areas before and after the “Wuhan lockdown” was announced was drawn.

As can be seen in Figure 7, before and after the lockdown, the average sentiment value of the texts posted by Wuhan netizens in Weibo was negative, indicating that Wuhan netizens held a negative attitude towards the epidemic regardless of the lockdown. After the lockdown was announced at 10:00 on 23 January 2020, the negative sentiment of Wuhan netizens was more serious.

However, in non-Wuhan areas, the average sentiment value of netizens about the epidemic was positive, which indicates that people in other regions generally held a positive attitude. When Wuhan announced the lockdown, people in non-Wuhan areas had more positive sentiments and better ability to resist rumors. 

## 6. Study 4 the Relationship between Rumor Spreading and Sentiment Polarity in the Context of COVID-19

### 6.1. Research Purposes

In study 4, 19,495 valid texts related to “COVID-19 rumors” and “epidemic refuting rumors” were used as the research object to explore the relationship between rumor propagation and emotional polarity in the context of COVID-19. 

### 6.2. Methods

Granger [77] gives the definition, inherent meaning, and test method of Granger causality testing. The Granger causality test is used to estimate whether past observations of X are useful for predicting Y [78].

### 6.3. Results

#### 6.3.1. Text Sentiment Vocabulary Index

Before discussing the causal relationship between rumor propagation and the emotional polarity of text expression in the context of COVID-19, we firstly need to train the Word2vec model and use the model to judge the emotional polarity of all the texts crawled.

Firstly, we randomly selected 10,000 pieces of data from the texts for artificial emotion assignment and recorded negative texts as −1, neutral texts as 0, and positive texts as 1. Then, we took 7000 texts from the Weibo sentiment test datasets as a training set for model training. Finally, the remaining 3000 texts were used as a test set for validation. Comparing the positive and negative emotion obtained by the model with the artificial emotion assignment, the results are shown in Table 9.

In this study, accuracy rate (*P*), recall rate (*R*), and F_1_ value (F_1_-score) were used as evaluation indexes. For the dichotomy problem, samples can be divided into true positive (*TP*), false positive (*FP*), false negative (*FN*), and true negative (*TN*) according to the real category and classifier predicted category. Then, the confusion matrix was constructed according to the calculation formula of the evaluation criteria.

accuracy rate: (5)$$P=\frac{TP}{TP+FP}$$

recall rate: (6)$$R=\frac{TP}{TP+FN}$$

F_1_ value: (7)$$F_{1}=\frac{2\cdot P\cdot R}{P+R}$$

Support vector machine (SVM), k-nearest neighbors (KNN), decision tree (DT), random forest (RF) and gradient boosting machine (GBM) are the more commonly used algorithms for text classification. Thus, we used the above methods to compare the results of text sentiment classification. From the perspective of classifiers, DT had best performance on *P*, *R*, and F_1_-score. SVM, KNN, RF, and GBM algorithms of positive texts were unsatisfactory, which means they might not be suitable for text classification in this study. 

#### 6.3.2. Correlation

SPSS 22.0 was used to explore the correlation between rumor and positive emotion, neutral emotion, and negative emotion, as shown in Table 10.

As can be seen from Table 10, these four variables are both positively correlated. 

#### 6.3.3. Granger Causality Test Results

In order to further explore the causal relationship between rumor and positive, negative, and neutral sentiments, we conducted a Granger causality test.

The following conclusions can be drawn from the Granger causality test results of bivariate time series in Table 11:

(1) There is a causal relationship between negative sentiments and rumor. Rumor will have an impact on positive sentiments, and neutral sentiments will have an impact on rumor, positive sentiments, and negative sentiments, indicating that the more negative people feel about COVID-19, the more likely rumors are generated.

(2) When a large number of rumors appear, negative sentiments will be generated. As for the neutral texts (such as the rumor refuting information) appearing in the network, some people will approve it and be optimistic about the rumor trend and the development of the epidemic, while others may be dissatisfied with the neutral texts (such as complaining about the delay in releasing the rumor refuting information).

## 7. Discussions

### 7.1. Classification and Prevalence of Weibo Rumors in the Context of COVID-19

In study 1, a Python crawler was used to crawl the Weibo texts from January 2020 to January 2021 with the keywords “COVID-19 rumors” and “epidemic refuting rumors”. After preprocessing the crawled texts, 21 topics and keywords under each topic were obtained via topic model analysis. By summarizing and classifying the topic of rumors, we can better explore the “popularity” of each topic and the relationship between rumors and sentiment polarity in the topic.

Li et al. [62] found that the susceptibility rate of ordinary people to rumors about the source and influence of the epidemic was low, while the susceptibility rate to rumors about the current situation and treatment of the epidemic was high. In addition, the anxiety of ordinary people and the degree of trust in various rumors could positively predict their willingness and behavior to respread rumors. The results of the study conducted by [62] were further confirmed in this study. As can be seen from Table 2, the most popular topic is “Netizens hope the official refute rumors”, followed by “Wuhan COVID-19 rumors”. The least popular topic was “Rumors cause social panic”, followed by “Internet spread rumors”. The popularity of “Netizens hope the official refute rumors” was much higher than other topics, indicating the netizens’ aversion to rumors and their desire for information about the real situation of the epidemic during the COVID-19 pandemic. In addition, as the region with the most severe COVID-19 outbreak at the beginning, Wuhan was the focus of discussion, so the prevalence of “Wuhan COVID-19 rumors” was also very high.

### 7.2. The Changes of Netizens’ Emotion before and after the Release of Rumor Refuting Information

Initial psychological studies show that human rumor-making and rumor-spreading behaviors are mutual mappings with inner emotion, which are driven by individual emotion such as anxiety, desire, and fear [34]. From topic popularity in study 1, we can see that “Netizens hope the official refute rumors” had the highest popularity. In order to further explore whether the rumor refuting information causes the change of Netizens’ emotion or not, in study 2, the rumor “71 cases of this epidemic have been confirmed in Qingdao” was taken as the research object, and sentiment analysis and time series analysis methods were used.

Kuang and Wu [49] proposed that one of the characteristics of the spread of rumors related to COVID-19 is that rumors always fluctuate with the development of the epidemic, but the overall life cycle of them is short. This conclusion was further confirmed in study 2. In study 2, the rumor that “71 cases of this epidemic have been confirmed in Qingdao” only lasted for four days in Weibo and changed significantly. The rumor study found that after the official released information to refute a rumor, the discussion on the rumor on Weibo first increased and then decreased. The main reason for this trend was that some netizens reposted and discussed the rumor in a large number of Weibo texts in the day after the official released information refuting a rumor, which led to the increase in the number of texts related to the rumor. Regarding this rumor, the proportion of positive texts was always higher than that of negative texts. Most netizens held positive attitudes and thought that this rumor was not credible.

There was a significant increase in positive sentiments after the official refuted the rumor. At the beginning of the rumor refutation, the proportion of negative texts was the highest. The reason was that after rumor refutation, some netizens would post their dissatisfaction with the rumor mongering and rumor mongers on Weibo. After netizens vented their dissatisfaction, the positive sentiments about the rumor gradually increased, and the final positive sentiments were higher than before the rumor was refuted. People thought that “fortunately, this was just a rumor and it was not true, the living environment was still safe”. This thought greatly reduced people’s panic and guaranteed people’s safety needs, so the positive sentiments about this rumor began to increase.

Finally, we found that the release of rumor refuting information can effectively suppress the spread of rumors, and after the release of rumor refuting information, it can increase people’s positive sentiments and decrease people’s negative sentiments.

### 7.3. Sentiment Changes of Netizens before and after the “Wuhan Lockdown” Event

In the early days of the COVID-19 outbreak, people’s panic and anxiety over the epidemic increased sharply, and the Chinese government announced the decision to “lockdown” Wuhan in order to prevent the further spread of the epidemic. [61] found that the parallel propagation of rumors and panic promoted each other, the generation of panic made the spread of rumors wider and have more influence, and the spread of rumor also intensified the spread of panic. In the study of the overall sentiment changes of netizens on rumor spreading before and after the “Wuhan Lockdown” event, we found that the discussion degree of Wuhan on Weibo was in a flat state, while people’s sentiment changes before and after the lockdown were large. Before the lockdown, because people in Wuhan could leave for other safer areas, people in other areas would have a sense of insecurity and panic. Therefore, negative sentiments were severe and negative texts appeared more frequently. When the lockdown was announced, people in Wuhan were temporarily “locked down”, while the safety needs of people in other areas were temporarily guaranteed, so their positive sentiments increased.

Then, we split the Weibo texts posted by Wuhan and non-Wuhan netizens and discussed the sentiment changes of people in Wuhan and non-Wuhan areas by calculating the average sentiment value. We found that regardless of whether Wuhan is locked down or not, the overall sentiment of Weibo texts in Wuhan was always in a negative state, while that in non-Wuhan areas was always in a positive state. When the lockdown was announced, people in Wuhan, which was at the center of the outbreak and restricted by policies such as home quarantine and no going-out, experienced a rapid increase in panic and unease and showed strong negative sentiments on Weibo. Dyer and Kolic [79] demonstrated their finding that as the pandemic intensifies, the proportion of words that appear in the set of Tweets posted in each country that indicate emotion diminishes over time. This indicates that the actual emotional response to the pandemic diminishes as the intensity of the pandemic increases, implying a psychophysical numbing effect. The relative severity of COVID-19 in Wuhan may have caused a psychophysical numbness among Wuhan people, resulting in a lack of positive sentiments. When people in non-Wuhan areas saw the news of the “Wuhan lockdown”, they thought their security needs were guaranteed to some extent, so after the lockdown, there were more positive sentiments in Weibo texts posted by netizens in other regions. 

### 7.4. The Relationship between Rumor Spreading and Sentiment Polarity

An important spread motivation of rumor mongers is emotional drive [52]. Internet rumors affect and even dominate the development direction of public opinions by mobilizing and stimulating group emotions, especially negative sentiment, and form sentiment public opinions [63]. Wang and Qiu [61] studied the propagation law of rumor and panic through mathematical methods and simulation and found that the parallel propagation of rumor and panic promotes each other. In study 4, we also obtained the same conclusion by calculating the sentiment index of the texts, the correlation, and the causality test. We found decision tree had the best performance when calculating the sentiment index of the texts. The decision tree classification technique is one of the most popular data mining techniques. A decision tree is a structure that includes a root node, branches, and leaf nodes. As decision trees mimic the human level thinking, it is simple to grab the data and make some good interpretations [80]. Decision tree classifiers obtain similar or better accuracy when compared with other classification methods [81]. Using decision tree models to describe research findings has the following advantages: they simplify complex relationships between input variables and target variables; they are easy to understand and interpret; they use a non-parametric approach without distributional assumptions; it is easy to handle missing values without needing to resort to imputation; it is easy to handle heavy skewed data without needing to resort to data transformation; they are robust regarding outliers [82]. From this study, we found that negative sentiments and rumors had mutual causality. In other words, in the context of COVID-19, the more negative sentiments people had, the more likely rumors were to be generated and spread. This was because when people were in panic, they needed to vent their panic and unease, while the cost of rumor mongering and rumor spreading was low in an anonymous Weibo environment, which provided a “hotbed” for the generation and spread of rumors. When rumors began to be generated and spread in large quantities, people would be plunged into deeper fear and their negative sentiments would be aggravated. This also proves that, in a sense, the generation and diffusion of network events and their public opinions are a process of sentiment mobilization [64]. Stella et al. [83] used MERCURIAL to analyze 101,767 tweets from Italy, the first country to react to the COVID-19 threat with a nationwide lockdown. They proposed that emotional polarization might therefore be a symptom of a severe lack of social consensus across Italian users in the early stages of the lockdown induced by COVID-19. In social psychology, social consensus is a self-built perception that the beliefs, feelings, and actions of others are analogous to one’s own. The panic that occurred when rumors spread during the COVID-19 pandemic may also have been related to the lack of social consensus.

In addition to negative sentiments, we also need to pay attention to the effect of neutral sentiments on the changes of people’s sentiments and rumors. As can be seen from Table 10, neutral sentiments had a positive correlation with rumor, positive sentiments, and negative sentiments. When netizens who held neutral sentiments towards the epidemic posted neutral texts (such as rumor refuting information) on Weibo, some may have said “fortunately, the rumor was not true and the epidemic was well controlled” after reading it, thus made positive comments and showed more positive sentiments. While some may have said that “rumor refuting information was not timely” or “rumor mongering and rumor mongers disturbed the social order”, which made these people release some negative comments and more negative sentiments. Whether the neutral texts became positive texts or negative texts when it was retransmitted would promote rumors to a certain extent and make the number of rumors rise.

## 8. Conclusions and Recommendations

### 8.1. Conclusions

Based on the above four studies, the following conclusions can be drawn:

(1) As for the types of rumors in the context of COVID-19, in study 1, we divided them into 21 topics by using the topic analysis model based on the texts obtained by the crawlers, and the most popular and widespread topic was the rumor “Netizens hope the official refute rumors”, indicating the public expectation for the official release of the true status of the epidemic. (2) In Study 2, the rumor that “71 cases of this epidemic have been confirmed in Qingdao” was taken as the research object to explore the sentiment changes of netizens before and after the release of rumor refuting information. After the rumor appeared, the official timely release of rumor refuting information reduced the rumor spreading, improved people’s positive sentiments, and increased people’s confidence in the fight against the epidemic. (3) In study 3, we studied the overall sentiment changes of netizens before and after the “Wuhan Lockdown” event and found that people’s sentiment changes before and after the lockdown were large. After taking measures for “high-risk” areas, people in “high-risk” areas experienced an increase in negative sentiments, while people in “non-high-risk” areas experienced an increase in positive sentiments. (4) In study 4, we discussed the relationship between rumor propagation and sentiment polarity. It is found that negative sentiments and rumors had mutual causality, and neutral sentiments were positively correlated with rumor, positive sentiments, and negative sentiments. The government and the media should guide the masses to view rumors rationally, so as to change the neutral sentiments to positive sentiments and reduce the generation of negative sentiments.

### 8.2. Recommendations

This epidemic was an unexpected public health event. In the face of rumors generated by emergencies, the government and people can correctly view and deal with rumors in the following three ways:

Firstly, the government should release true and reliable information on relevant online platforms in a timely manner, refute rumors in a timely manner, and guide people in the correct direction of public opinions. The government also should open information complaint and report platforms, establish a perfect and sound supervision and complaint system, play the role of social supervision fully, and detect and block the spread of rumors in timely manner.

Secondly, the government should improve the ability to identify and block Internet rumors, purify the network environment, and reduce the generation and spread of rumors.

Finally, the public need to improve their ability to identify rumors, “not believe rumors”, and “not spread rumors” and consciously cooperate with the government and other institutions to create a good public opinions environment for the society.

## Figures and Tables

**Figure 1 healthcare-09-01275-f001:**
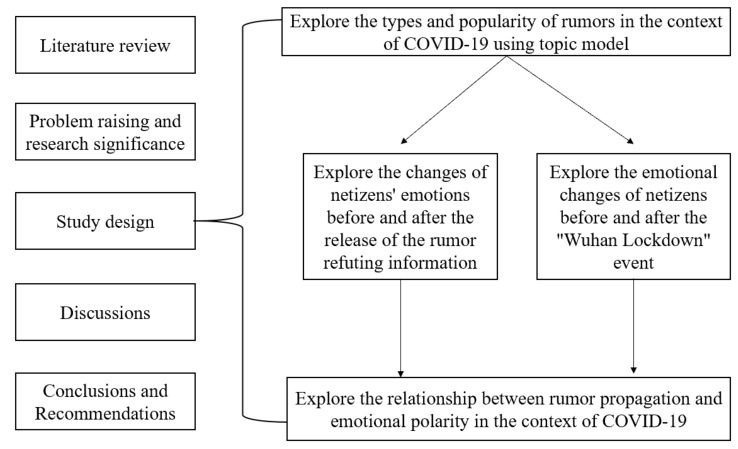
Research framework.

**Figure 2 healthcare-09-01275-f002:**
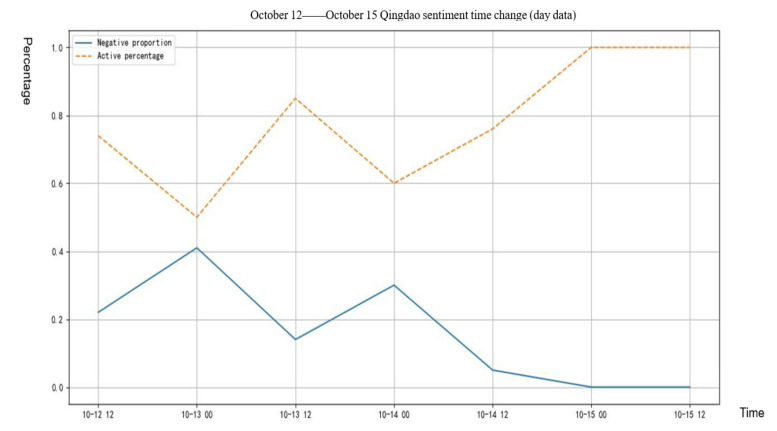
Positive and negative text percentage change.

**Figure 3 healthcare-09-01275-f003:**
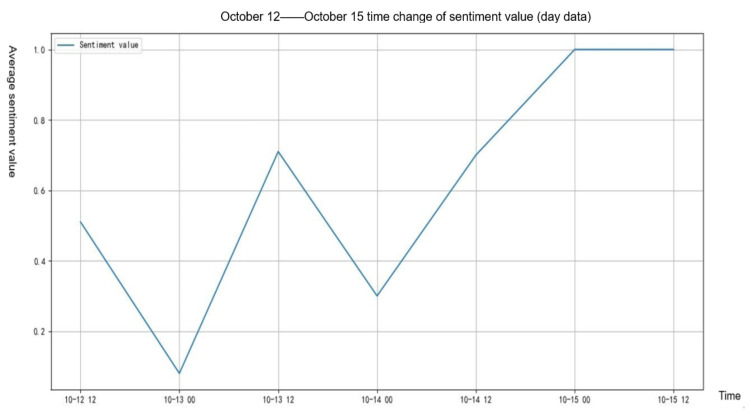
The change of daily average sentiment before and after refuting rumors.

**Figure 4 healthcare-09-01275-f004:**
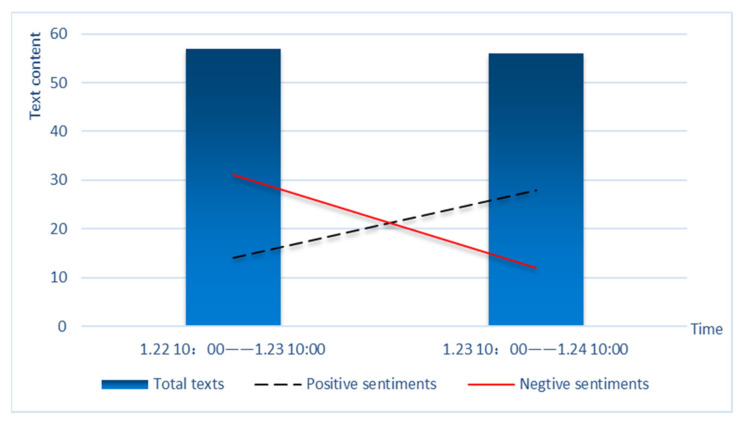
Sentiment changes of netizens before and after Wuhan lockdown.

**Figure 5 healthcare-09-01275-f005:**
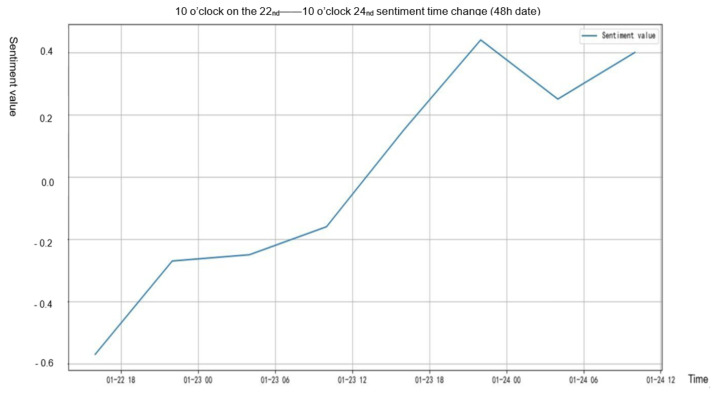
Sentiment value changes of “Wuhan Lockdown” event.

**Figure 6 healthcare-09-01275-f006:**
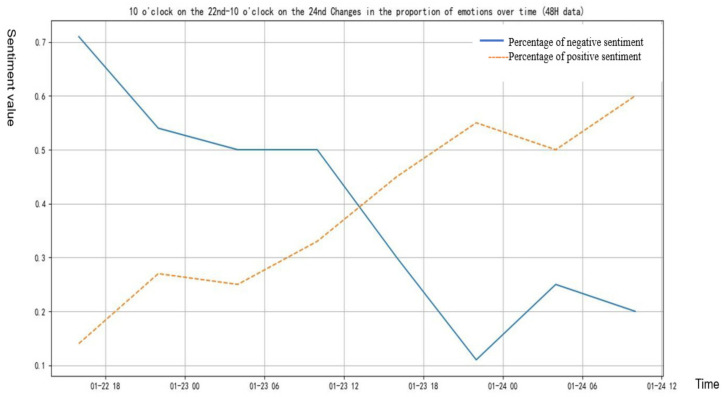
Changes of positive and negative texts before and after “Wuhan Lockdown”.

**Figure 7 healthcare-09-01275-f007:**
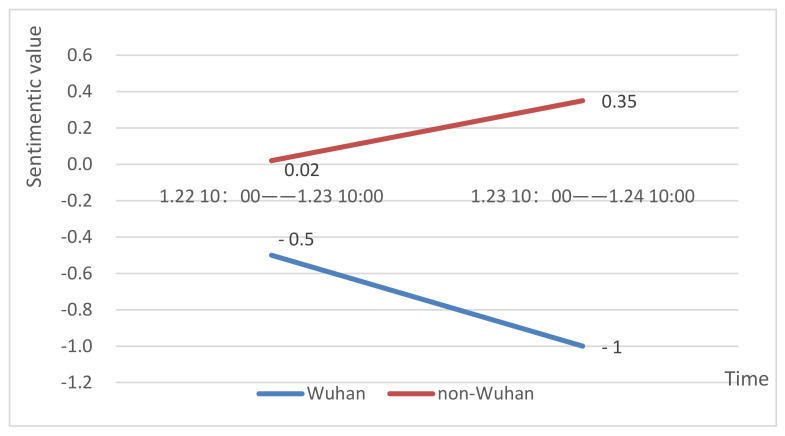
Sentiment changes in Wuhan and non-Wuhan areas.

**Table 1 healthcare-09-01275-t001:** Weibo topic keywords.

	Topic	Keywords
0	COVID-19 rumors in Wuhan	hospital, COVID-19, Wuhan, school opens, infections, patients
1	Epidemic prevention and control	prevention and control, work, epidemic prevention, complete, mask, protection
2	British 5 g base station spreads COVID-19 virus	5 g, British, spread, COVID-19 virus, signal, base station
3	Rumors cause social panic	spread, cause, society, don’t believe, netizens, false
4	Internet spread rumors	video, scary, male, spread, COVID-19 virus, Hubei
5	Rumor refutation all over China	confirmed cases, COVID-19, Internet transmission, Cases, Beijing, new cases
6	COVID-19’s origin	COVID-19 virus, spread, infection, experts, evidence, research
7	Market enterprise downtime	novel, COVID-19 virus, pneumonia, infection, prevention and control, market
8	World COVID-19	China, country, COVID-19, virus, world, global
9	Nucleic acid detection of suspected personnel	detection, nucleic acid, personnel, positive, isolate, negative
10	The public security bureau detained rumor mongers	spread, police, law, public security organs, rumor, release
11	Weibo users release rumors	video, Weibo, Zhong Nanshan, academician, refuting rumors, disinfection
12	Company rumor refutation	Beijing, news, company, related, group, official
13	COVID-19 vaccination	The United States, vaccines, COVID-19 virus, China, Trump, vaccination
14	Rational treatment of rumors	Spreading rumors, disbelieving rumors, making rumors, prevention and control, information, release
15	Series of rumors of tea and garlic wipe out COVID-19 virus	COVID-19, prevention, mask, patients, treatment, infection
16	Netizens hope the official refute rumors	true, start a rumor, hope, come on, Weibo, period
17	Nine latest rumors	Qingdao, diagnosis, data, official, circle of friends, latest
18	COVID-19 rumors lead to many deaths	death, COVID-19, report, article, India, Iran
19	Netizens spread false information	information, spread, WeChat group, news, fake, netizens
20	Internet rumor refutation	network, platform, Internet, China, joint, release

**Table 2 healthcare-09-01275-t002:** Popularity of each topic (volume of text).

Topic	Volume of Text (Popularity)
Netizens hope the official refute rumors	2278
COVID-19 rumors in Wuhan	1289
Series of rumors of tea and garlic wipe out COVID-19 virus	1277
World COVID-19	1275
COVID-19’s origin	1227
Rational treatment of rumors	1181
COVID-19 vaccination	1125
Rumor refutation all over China	1002
Nine latest rumors	997
The public security bureau detained rumor mongers	933
Netizens spread false information	860
Epidemic prevention and control	799
British 5 g base station spreads COVID-19 virus	693
Nucleic acid detection of suspected personnel	672
Weibo users release rumors	655
Market enterprise downtime	626
Internet rumor refutation	597
COVID-19 rumors lead to many deaths	582
Company rumor refutation	567
Internet spread rumors	453
Rumors cause social panic	407

**Table 3 healthcare-09-01275-t003:** Linear regression of the proportion of positive texts before and after refuting rumors.

	B	Standard Error	Beta	T	Significance
Constant	0.555	0.178		3.122	0.089
Time	0.085	0.065	0.679	1.309	0.321

**Table 4 healthcare-09-01275-t004:** Linear regression of the proportion of negative texts before and after refuting rumors.

	B	Standard Error	Beta	T	Significance
Constant	0.385	0.111		3.477	0.074
Time	−0.085	0.040	−0.830	−2.102	0.170

**Table 5 healthcare-09-01275-t005:** Linear regression of sentiment value before and after refuting rumors.

	B	Standard Error	Beta	T	Significance
Constant	−0.86	0.226		−0.379	0.720
Time	0.150	0.051	0.798	2.964	0.031

**Table 6 healthcare-09-01275-t006:** Linear regression of positive text proportion of “Wuhan Lockdown”.

	B	Standard Error	Beta	T	Significance
Constant	0.590	0.064		9.179	<0.001
Time	−0.010	0.013	−0.315	−0.813	0.447

**Table 7 healthcare-09-01275-t007:** Linear regression of negative text proportion of “Wuhan Lockdown”.

	B	Standard Error	Beta	T	Significance
Constant	0.731	0.072		10.121	<0.001
Time	−0.076	0.014	−0.909	−5.340	0.002

**Table 8 healthcare-09-01275-t008:** Linear regression of sentiment value of “Wuhan Lockdown”.

	B	Standard Error	Beta	T	Significance
Constant	−0.634	0.101		−6.244	<0.001
Time	0.140	0.020	0.944	6.979	0.000

**Table 9 healthcare-09-01275-t009:** Text sentiment vocabulary index.

		Precision	Recall	F_1_-Score	Support
Support Vector Machine (SVM)	negative texts	0.66	0.73	0.69	958
	neutral texts	0.77	0.90	0.83	1501
	positive texts	0.77	0.26	0.39	541
	avg/total	0.74	0.73	0.71	3000
K-Nearest Neighbors (KNN)	negative texts	0.74	0.78	0.76	843
	neutral texts	0.88	0.92	0.90	1872
	positive texts	0.79	0.40	0.53	285
	avg/total	0.80	0.70	0.73	3000
Decision Tree (DT)	negative texts	0.92	0.92	0.92	878
	neutral texts	0.96	0.97	0.96	1833
	positive texts	0.86	0.82	0.84	289
	avg/total	0.91	0.90	0.91	3000
Random Forest (RF)	negative texts	0.69	0.63	0.66	853
	neutral texts	0.77	0.93	0.84	1848
	positive texts	0.00	0.00	0.00	299
	avg/total	0.49	0.52	0.50	3000
Gradient Boosting Machine (GBM)	negative texts	0.76	0.74	0.75	892
	neutral texts	0.84	0.93	0.88	1825
	positive texts	0.80	0.33	0.46	283
	avg/total	0.80	0.67	0.70	3000

**Table 10 healthcare-09-01275-t010:** Rumors, positive emotions, neutral emotions, and negative emotions number correlation table.

	Rumors	Negative Sentiments	Neutral Sentiments	Positive Sentiments
Rumors	1			
Negative sentiments	0.853 **	1		
Neutral sentiments	0.962 **	0.698 **	1	
Positive sentiments	0.829 **	0.845 **	0.836 **	1

Note: ** at level 0.01 (double tail), the correlation is significant.

**Table 11 healthcare-09-01275-t011:** Granger causality test of bivariate time series.

Null Hypothesis	*p*-Value	Conclusion
Negative sentiments have no causal effect on rumors Rumors have no causal effect on negative sentiments	0.032 0.022	reject reject
Positive sentiments have no causal effect on rumors Rumors have no causal effect on positive sentiments	0.072 0.025	accept reject
Neutral sentiments have no causal effect on rumors Rumors have no causal effect on neutral sentiments	0.007 0.165	reject accept
Positive sentiments have no causal effect on negative sentiments Negative sentiments have no causal effect on positive sentiments	0.944 0.468	accept accept
Neutral sentiments have no causal effect on negative sentiments Negative sentiments have no causal effect on neutral sentiments	0.006 0.134	reject accept
Neutral sentiments have no causal effect on positive sentiments Positive sentiments have no causal effect on neutral sentiments	0.008 0.087	reject accept

## Data Availability

The data presented in this study are available on request from the corresponding author.

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
