# Peer review of "Sentiment Analysis of Rumor Spread Amid COVID-19: Based on Weibo Text"

_healthcare, 2021, doi:10.3390/healthcare9101275_

Round 1

Reviewer 1 Report

This paper analyzes changes in the sentiments of users of a social network after the spread of rumors about covid-19. The subject of the paper is interesting, however, the part corresponding to sentiment analysis methods and time series presents serious deficiencies that need to be solved. I recommend the authors to read the following papers that may help them to improve their work:

  • Dang, C., Moreno-García, M.N., De la Prieta, F. “Sentiment Analysis Based on Deep Learning: A Comparative Study”, Electronics, 9 (3), 483, 29 pages, 2020.
  • Dang, C., Moreno-García, M.N., De la Prieta, F. “Hybrid Deep Learning Models for Sentiment Analysis”, Complexity, vol. 2021, Article ID 9986920, 16 pages, 2021.

Aspects to be improved are detailed below.

  1. The paper lacks a literature review regarding machine learning methods that can be applied to sentiment analysis.
  2. Some statements related to scientific concepts are incorrect.
    • Statement “Word vectors is also known as Word embedding” is not true because word vectors can also refer to vectors containing the frequency of words (absolute, binary, TF-IDF...). Moreover, word embedding is not used in this work but TF-IDF, unless it is being used to weight word embeddings, which is not clearly specified.
    • The definition “supervised machine learning methods were used to train the classified texts, the test texts were used to test training model and to get the classification results of the model” is confusing and grammatically incorrect. There are much more appropriate definitions of these methods in the literature.
    • The term "verifying" in "7,000 texts from the artificial emotion assignment for model training. Finally, the left 3,000 texts were taken for verifying" is not appropriate. The data are divided into training and test sets and the test set is used for validation of the classification model.
  3. The description of the procedure used for sentiment analysis is very sketchy.
    • There is no indication of how "emotional words modified by dependency" and " emotional words not modified by dependency " are identified and counted, nor how their SentimentScore is calculated. Is the sentiment calculated for each word? Normally it is calculated for text containing several words.
    • It is not indicated how the weight values of emotion in eq. 2 are obtained.
    • In two consecutive equations (1 and 2) the variable “m” appears with two different meanings.
    • It is not clear at what point and for what purpose the Word2vec model is used. It appears that equations 1 and 2 are used to label the sentences of the training and test sets so that the sentiment analysis method used can be applied and validated later, but this is not clear from the manuscript.
    • The authors do not specify which machine learning method they used for sentiment analysis.
    • No details are given on the time series analysis method or how it is applied.
  4. The classification results (precision, recall...) are not good. Perhaps this is due to the fact that an appropriate machine learning method has not been used. Currently deep learning algorithms are giving very good results.

Reviewer 2 Report

The authors produce an interesting language inquiry of Weibo messages in temporal proximity of the Wuhan lockdown. By quantifying sentiment scores of text messages and their spread over time, the authors identify a set of different results, the most important being a correlation between negative affect and increased spreading rate of posts.

The manuscript is mostly well written and rich in relevant references. The narrative should be made more cohesive and an ambiguity between emotions and sentiment should be addressed. A Figure should be corrected and the presentation of some passages could use a few more references relative to COVID-19 and emotional profiling. Nonetheless, the authors should be praised for their multidisciplinary approach to text analysis and for producing results of interest in view of the current pandemic. For these reasons, I recommend major revisions.

---

The Introduction is nicely written and rich in different references. There are some issues with a fragmented flow of the narrative presenting the scope of the paper. I hope the comments down below can help in restructuring this part to make the study more appealing the readers.

In the Introduction, the references relative to the definition of “rumour” are relevant but out of date. The first 2 paragraphs would benefit from a rewriting with more recent references. A suggestion could be mentioning the “dark side of information proliferation”, see Hills, Persp. Psych. Sci., (2019). Some of the sentences from Section 1.2 could be frontloaded at the beginning of the Introduction.

In Section 1.1.3 there is ambiguity between “emotion” and “sentiment”. Sentiment can be negative, neutral or positive, emotions are considerably more nuanced, e.g. sadness can be both neutral and negative according to other emotional states (e.g. melancholia vs desperation). It’s also important to underline that sentiment can take into account only the positive/negative dimension of affect, whereas emotions can include also other dimensions like arousal, e.g. both depression and anxiety are negative emotional states but depression inhibits arousal whereas anxiety boosts it. For a review about emotional profiling in social media contexts, which is a key aspect of this paper, I would recommend for the authors to reference Stella, Topics in Cog Sci (2021). This reference could be used in Section 1.3.3 when mentioning that social media can be a way for people to express their own emotions.

1.3.1: big data -> Big Data. In the first sentence, both the above references could be used to strengthen the fact that social media drive people’s opinions online and are key for knowledge search, thus making digital platforms crucial for the spread of rumours.

Section 1.3.2 and 1.3.3 are very informative sections, yet they feel disconnected. I would recommend for the authors to merge these sections with the above ones and frontload the argument of detecting rumour spreading in social media during major health crises.

1.4: present study -> Present Study

3.1 -> Please add a reference to the “Python crawling method” mentioned for gathering the data. Which library was used?

3.2.1 -> How did the authors identify 21 as the correct number for topic detection? Did they use any metrics like coherence or a priori knowledge?

In the Table 1 -> In Table 1

4.2.2 – Please add a reference to sentiment analysis. Where did sentiment scores come from? Did they come from VADER or from the Valence-Arousal-Dominance dataset by Mohammad and Turkey 2013?

Please reformat Equation 7 with shorter notations.

zigzag -> noisy

Table 6 – 8 : What does a significance of 0.000 mean?

Please correct Figure 7.

study 1 -> Study 1

Please correct it everywhere in the Discussion: The current methods focus on sentiment, which is a subcomponent of emotions. Whereas there is “negative emotion”, the correct wording should be “negative affect” or “negative sentiment”. Please see Dyer and Kolic, APN (2020) and Stella et al., BDCC, (2020) for  examples of emotional profiling models at work over people’s reactions to a national lockdown after COVID-19 was declared a pandemic. These or other references might enrich the discussion at the end, which reads quite succinct and lacks references to other recent approaches with affective profiling about COVID-19.

Round 2

Reviewer 1 Report

The authors have not addressed most of the observations made in the first review.

  1. The requirement to include in the literature review and overview of machine learning methods used for sentiment analysis has not been met. Only a few lines on generalities have been added and deep learning techniques are mentioned directly without having previously commented on any work where other machine learning techniques are applied.
  2. The authors state that they have used TF-IDF from Word2Vec, however TF-IDF is not included in this model. Sometimes it is used in a preprocessing step as a weighting strategy. The vectors produced by Word2Vect are based on unsupervised learning and take context into account whereas TF-IDF model does not consider the context. The context is formed by the words surrounding the target word.
  3. The new paragraph introduced in section 4.2.2 (“Finally, we used supervised machine learning…”) is an accumulation of inaccuracies as well as incomplete, confusing, and misplaced information. First, the authors still do not specify which machine learning method they have used. In addition, within the paragraph a definition of TF-IDF appears that should have been put earlier when TF-IDF was first mentioned. Moreover, they talk about word vectors in that paragraph and in the preceding one without specifying which vectors are the input to the Word2Vec model and which are output vectors. They do not say which particular word2Vec model was used. Finally, it is said that word2Vec has been used to train the texts, but it is not indicated how the sentiment classification is done. The word2Vec model produces probability vectors for each word with respect to the other words but does not classify the sentiments into positive or negative. How is the classification done?
  4. One of the major weaknesses of the work, pointed out in the previous review, is related to the poor results and the lack of comparison with other methods that could improve them. This issue is ignored by the authors and relegated to future work.
  5. English is poor, especially in the text that has been added.

Reviewer 2 Report

The authors addressed all my points and the overall quality of the manuscript improved considerably. For this reason, I recommend accepting this paper for publication.
